# New Hydraulic Sensor for Distributed and Automated Displacement Measurements with Temperature Compensation System

**DOI:** 10.3390/s21144678

**Published:** 2021-07-08

**Authors:** Łukasz Bednarski, Rafał Sieńko, Piotr Kanty, Tomasz Howiacki

**Affiliations:** 1Department of Mechanics and Vibroacoustics, Faculty of Mechanical Engineering and Robotics, AGH University of Science and Technology in Krakow, Mickiewicza 30, 30-059 Krakow, Poland; lukaszb@agh.edu.pl; 2Reinforced Concrete Structures Division, Faculty of Civil Engineering, Cracow University of Technology, Warszawska 24, 31-155 Krakow, Poland; rafal.sienko@pk.edu.pl; 3Menard Polska Sp. z o.o., Powązkowska 44c, 01-797 Warsaw, Poland; pkanty@menard.pl; 4SHM System Sp. z o.o., Sp. kom., Libertów, ul. Jana Pawła II 82A, 30-444 Kraków, Poland

**Keywords:** hydraulic sensor, displacements, settlements, measurements, temperature, thermal compensation, embankment, laboratory, distributed sensing, DFOS

## Abstract

Structural health monitoring (SHM) is a challenging task, especially in the context of ground and geotechnical structures. They are characterized by a set of random mechanical parameters, depending on the location but also changing with external conditions (such as humidity or temperature) over time. Theoretical predictions and results of numerical simulations are, therefore, considerably uncertain. On the other hand, measurements aimed at improving construction and operation of such structures are very often performed only in selected points, which significantly increases the risk of data misinterpretation. Reliable measurement data related to structural condition are of the great importance because they allow for improvement of work quality but also reduce construction time and, thereby, save money. That is why scientists and engineers are still searching for new measurement solutions to overcome existing limitations. The purpose of the study is to present the design and practical application of a new hydraulic sensor dedicated to vertical displacement sensing. The novelty of the presented solution lies in several features, including the possibility of performing automatic measurements and compensating the results due to temperature effects. The article describes the sensor’s design, including the concept of a thermal compensation system and example results from laboratory tests, where the sensor’s performance was investigated in a dual-zone thermal chamber. Finally, the sensor was installed within the field conditions under an embankment constructed above the improved substrate. Example results verified by reference distributed fiber optic technique are presented and discussed hereafter, raising high prospects in the context of possible structural health monitoring applications of the new solution.

## 1. Introduction

Structural health monitoring SHM [1] is a process of obtaining objective knowledge about the technical condition of the structure over time (long-term monitoring [2]) and, thus, to optimize decision making [3,4]. These decisions influence many important aspects, including improving safety and quality [5], reducing time of works, and providing financial savings, both in the short term (construction) and over the entire lifecycle of the structure [6].

Although the origin of monitoring process and development of sensing solutions usually comes from the aerospace industry [7,8], it has become more and more useful in civil engineering [9] and geotechnical [10] applications. The idea of measurements remains the same. However, to utilize all the benefits coming from the monitoring (SHM), the measuring tools (sensors) should be modified, adjusted, or designed from scratch to meet the specific requirements of the construction sector. In geotechnics, the need for monitoring is highlighted and described in the projects of new design codes (prEN1997-1 and prEN1997-3), which are still under development and will be released by CEN after 2023.

Currently, the most common measurements are those made using spot sensors based on different physical principles, e.g., inductive or vibrating wire [11,12]. Automatic measurements with spot sensors are especially useful for relatively homogeneous and continuous materials (such as steel, plastics, composite, timber), where results in selected points could be reasonably extrapolated for assessment of the structural behavior within other parts of the element. However, using spot measurements, it is impossible to detect local damages [13], disturbances, or discontinuities, such as cracks in concrete [14] or ground fractures. Randomness of physical and mechanical parameters of such materials depending on location significantly influences reliability and certainty of the obtained results. In the case of concrete structures, apart from cracks, we can observe influence of pores, surface and internal imperfections, or the random presence of aggregate grains changing the stiffness over length. On the other hand, ground parameters depend on external conditions (e.g., humidity, water level, temperature), and they are variable not only by location but are also changing over time.

This is the reason why, presently, scientists and engineers are working to improve existing or develop new measurement solutions based on distributed sensing. Unlike spot sensors, distributed techniques allow the measurements of selected physical quantities to be performed in continuous way along the entire structural length. This provides the ability to directly locate and identify any anomalies and, thus, significantly improve the quality of analysis and assessment of technical condition of the monitored structure. Figure 1 summarizes the main advantage of the distributed approach over conventional spot solutions with the example of settlement measurements under an embankment constructed on the substrate strengthened with concrete columns.

Distributed sensing could be realized based on different physical principles. One of the most promising is distributed fiber optic sensing (DFOS) involving light scatterings: Rayleigh [15], Brillouin [16], or Raman [17]. Despite many laboratory and industrial proven applications [18], one of the main limitations is still the high cost of optical dataloggers (reflectometer, interrogators) required for data acquisition. Another approach met in practice for distributed sensing is based on hydraulic systems, which are much cheaper in comparison with optical measurements.

According to the state-of-the-art review and authors’ own experiences, there are only a few automatic hydraulic systems dedicated for distributed vertical displacement measurements for civil engineering and geotechnical purposes. All of them share the advantages of not only distributed sensing but also high accuracy, large measuring range, excellent resistance to harsh environmental conditions, and low costs of both the sensors and data acquisition systems. Example of possible proven applications under earthen dike, sand embankment, and railroad embankment are presented in Lhotzky and Frield [19].

Hydraulic systems for distributed measurements were checked in the context of local damage detection, which is one of the main requirements for structural health monitoring systems and the basic advantage of distributed sensing as well. The main principle of operation is based on the use of a special tube in which a suitable liquid is pumped along the entire length. The liquid is stopped at selected positions (with defined spatial resolution), and after stabilization, the liquid column pressure is measured by the transducer located at the end of the tube—Figure 2. This pressure is further processed to obtain height profile (shape of the sensor in vertical plane) over the entire measuring length. The hydrostatic pressure equation is used for this purpose. The spatial resolution (number of measuring positions along the length) should be individually adjusted depending on the load scheme, as well as a type of predicted deformations.

Despite many obvious advantages described above on the example of real applications, none of the known hydraulic systems for distributed displacement measurements allow for the construction of a remote and automated structural health monitoring systems. The main limitations of existing solutions are as follows:the need for at least one operator to handle the measurements;significant influence of temperature changes on measurement accuracy. Temperature, due to the thermal expansion of the system components (tube, liquid, air), affects pressure indications, causing the measurement errors;the necessity of long-term drying of the hydraulic tube with the use of expensive technical gases after each single measuring session;the need for a dry gas reserve and its regular refilling for the continuity of the sensor’s operation;the frequency of necessary services and the amount of technical gases consumed are directly proportional to the length of the measuring line and the frequency of measurements. This dependence makes the applications in places with significant variability of vertical displacements over time (e.g., bridges) uneconomical.

Design and implementation of a new hydraulic sensor and automated measurement system, overcoming the existing limitations, are described hereafter in the article. The sensor’s performance, including the effectiveness of the temperature compensation system was verified in laboratory conditions within a special dual-zone thermal chamber. Finally, the sensor was successfully installed in difficult geotechnical conditions during construction of the road embankment. Lessons learned during this project are also described.

Once again, it is worth underlining for clarity of the message that many hydraulic measurement systems used commonly in practice are based on multiple spot sensors (Figure 3a) or a single sensor (probe) moved by hand to different measuring locations (Figure 3b). The further-described solution for geometrically continuous measurements should not be confused with this quasi-distributed systems or their manual equivalents.

## 2. New Hydraulic Displacement Sensor

### 2.1. Basic Operation Principle

The main measuring element of this new hydraulic sensor (called themSHM Profiler) is a tube filled with special liquid separated by an air bubble with appropriate length *L_b_*. The main measuring tube with additional equipment, including electronic and sensing devices, forms a closed system (loop) in which the liquid and air bubble can be pumped back and forth using the precise stepper pump—Figure 4a. This is one of the basic requirements for creating an automated system that does not involve human operation.

The determination of the sensor’s profile (shape) over the entire considered length *L* is based on the measurements of the hydrostatic pressure in position *i* generated by a column *h_i_* of the liquid inside the tube relative to a reference level (location of the pressure transducer). This rule is illustrated in Figure 2. However, in the presented approach, the hydraulic system is designed in form of a closed loop, enabling the measurements to be repeated automatically. The liquid with bubble is moved along the measuring tube at a predefined distance (spatial resolution *r*), resulting from the number of steps done by the stepper pump. This distance was checked and calibrated in laboratory for a specific type of pump, tube diameter, and type of liquid. After each number of steps, the liquid with bubble is stopped in a specific position *i* to take the pressure measurement. Air bubble detectors indicate the beginning and the end of the actual measurement length. When the start point is identified, the pump step distance is then added to it. When the end point is identified, the procedure is repeated from the other side.

As the relationship between the height of the liquid column and the measured pressure is linear (directly proportional—Equation (1)), the actual shape of the hydraulic sensor is determined directly during the measurement session using the following, basic equation:(1)$$h\left(x_{i}\right)=f\left(p\left(x_{i}\right)\right)=\frac{p\left(x_{i}\right)}{g\times \rho \left(T\right)}\;$$
where *g* is gravitational acceleration in the place of measurements and *ρ*(*T*) is the liquid density depending on actual temperature *T*.

It is worth underlining that the air bubble moves along the tube together with the liquid, separating two liquid columns. Because of this, it is possible to make measurements with two independent stationary pressure transducers, located on both sides of the tube in the hydraulic system. This approach increases measurement accuracy by averaging the results from two pressure transducers, allowing us to avoid gross errors and improve the system’s reliability. However, the appropriate length of the air bubble must be ensured to not transfer the pressure from one liquid column to another. The longer the air bubble, the lower the influence of one liquid column to another. Although the bubble itself can be compressed, it is possible to find a length that does not affect the indications of two independent pressure transducers. The length of air bubble depends on maximum (predicted) range of vertical displacements, and it was established experimentally.

Integrity of the air bubble is controlled by bubble detectors located at the beginning and at the end of the measuring line. Details of the components of hydraulic system are also presented and discussed in Section 2.4.

The difference between the height profiles (shapes) of the sensor, determined in the actual and reference measurement session, provides the sought vertical displacement Δ. An example of the spatial visualization of this principle is presented in Figure 4b, for which an adequate equation is as follows:(2)$$\Delta \left(x_{i}\right)=h_{act}\left(x_{i}\right)-h_{ref}\left(x_{i}\right)$$

### 2.2. Design Issues

Although the general operation principle of the sensor is trivial and based on a well-known physical law, there are many aspects and technical details that influence the final accuracy of the measurement and, thus, the suitability of the proposed solution for practical applications. The article describes only selected issues in detail, including displacement accuracy or efficiency of thermal compensation system (TCS)—see Section 3. However, a very wide range of laboratory tests has been carried out in order to gain new knowledge and overcome existing limitations. Table 1 summarizes explored issues in order to give a brief overview of the challenges involved in developing a new hydraulic sensor and automatic measurement system. These seemingly simple issues have caused no analogous practicable solution to be developed to date.

### 2.3. Sensor Design

During different types of laboratory studies, briefly summarized in Table 1, many prototype versions of the new sensor were investigated to choose the optimal solution. The main criteria during the design stage were:sensor’s performance in difficult geotechnical conditions;required accuracy and effectiveness of the thermal compensation system;feasibility and ease of establishing a production line;low costs of the sensor.

The prototype version of the sensor, which was finally demonstrated within three field installations (including the road embankment described in Section 4) consisted of:main tube filled with measuring liquid separated with air bubble. The liquid was pumped with a specified spatial resolution in order to perform the pressure measurements when the height (shape of the sensor) varied along the length;compensation tube filled the same way as the main tube; however, the liquid with air bubble was fixed to compensate for all effects affecting pressure measurements, which were not related to change in height (shape) of the sensor. The readings from the compensation tube were subtracted from the main tube in each measuring position;a set of six tubes filled with the liquid circulating in a closed circuit, the purpose of which was to balance (equalize) the temperature over the entire sensor length. Constant temperature in length domain (not necessarily in time domain) is one of the most important factors providing required accuracy;external cover, whose main aim was to protect the sensor’s components against mechanical damages and aggressive and harsh environmental conditions, as well as facilitate the transportation and installation process.

Cross section of the prototype version of the new hydraulic sensor is presented in Figure 5a, while its spatial visualization is tin Figure 5b. Example general view and the close-up to the cross section of he ready-to-install product are shown in Figure 6a,b respectively.

### 2.4. Hydraulic System (Data-Logger)

Design and construction of the sensor are of great importance to provide appropriate accuracy, performance, and resistance to external conditions. However, another important aspect is the hydraulic system for sensor operation. The main objective of such a system is to enable remote and automatic measurements by means of electronic devices and informatic software. Furthermore, the data logger should be compatible with any SHM Profiler sensor so that the measurements can be taken periodically on different projects. All electronics and additional accessories required for correct measurements should be closed in an ergonomic and transportable case that can be moved from place to place (e.g., from one construction site to another).

Automated measurements are possible due to the application within hydraulic system (Figure 7) pressure and temperature transducers, air bubble detectors, and reservoirs with the liquids, as well as electro-waves. In addition, it was necessary to design calibration loops increasing the accuracy by eliminating constant error, as well as air bubble storages that allowed measurements to be taken while the liquid was being pumped back and forth (the bubble could not pass through the pump components due to the risk of damage). The measurement session was executed by electronics controlled by the software, in which a number of measurement parameters could be predefined by the user, including spatial resolution, length of air bubble, or calibration constants.

The system described above was designed (Figure 8a), constructed, and enclosed in a compact case with dimensions of 800 mm × 600 mm × 450 mm (Figure 8b), which was then checked against harsh conditions within three demo installations. It is worth noticing that the ability of connection between the data-logger and the sensor was possible due to elaboration of the appropriate connection (see also line 10 in Table 1), which did not destroy the air bubble during the pumping.

## 3. Laboratory Studies

This section is focused on two key aspects, including accuracy of displacement profile measurements, as well as the effectiveness of the thermal compensation system (TCS), whose main aim is to balance (equalize) the temperature field along the entire sensor length. It should be emphasized that the research described hereafter is only a short extract from the entire study, which covered much broader issues, presented briefly in Table 1.

### 3.1. Displacements Accuracy

Laboratory tests were performed using a special measurement station in form of a stiff, steel framework (Figure 9a) where sensor specimens were stabilized with appropriate mounting blocks—Figure 9b. To control all the experiments and analyze the final accuracy of the new hydraulic sensor, reference techniques were applied, including both inductive LVDT displacement sensors with accuracy of ± 0.05% and linear encoders with accuracy ± 0.02% of their full scale—Figure 9c. Encoders were used to analyze sensor performance in the range of large displacements, simulating the behavior within the landslide areas.

Sensor displacements were controlled using linear tables with a screw mechanism, allowing for any shape configuration in both small and large displacement ranges. In order to increase the final accuracy, it was possible to use a pressure transducer with a measuring range adapted to the expected values of displacements. The measurement session was performed with defined spatial resolution equal to 10 cm. Constant temperature was provided during research by a thermal chamber, in which the measuring station was located. After processing the raw pressure data [kPa], calculated displacements [mm] were compared with the results from reference gauges *h_ref,ij_* located in selected checkpoints along the sensor specimen. The differences in heights (levels) calculated between two specified checkpoints (*x_i_, x_j_*) were then used to calculate the accuracy of the sensor *re* relative to its full measurement scale *fs*—Equation (3):(3)$$re_{ij}=\frac{h_{ref,ij}-\left(h\left(x_{i}\right)–h\left(x_{j}\right)\right)}{fs}\times 100\%$$

Figure 10a shows a theoretical scheme representing the above considerations, while Figure 10b represents graphically the results coming from six measurement sessions from one example series. Corresponding results are summarized in Table 2, showing very good accuracy of the proposed hydraulic system under laboratory conditions, especially including a constant temperature environment.

Data were processed using two algorithms:Based on calibration loops located at the beginning and at the end of the measurement length, whose aim was to eliminate the constant error related to actual conditions during measurement session;Compensating for the positioning error resulting from the limited accuracy of applied stepper pump. Based on local approximation approach [22], it was possible to estimate the sought values of displacement at fixed checkpoints, no matter the real measurement positions are. This algorithm is important not only due to the pump precision but also due to possible large displacements significantly changing the geometry of the sensor and, thus, the locations of measuring points within horizontal axis.

The mean error for uncorrected data was approximately equal to 1% of the full displacement scale. However, application of elaborated algorithms allowed us to improve the accuracy by up to 0.1% with standard deviation equal also to 0.1%.

### 3.2. Thermal Compensation System (TCS)

Another type of laboratory research was focused on thermal sensitivity and the possibility of efficient thermal compensation. Performed analytical analysis and elaborated algorithms indicated that the main requirement for accurate calculations is to ensure the constant temperature over the entire length during asingle measurement session. Mean temperature could change in time but could not vary along the length by more than ±1 °C. Temperature differences up to a few or even several degrees Celsius are common in practice, e.g., within the bridges (northern and southern parts) or geotechnical embankments (the temperature depended on external conditions near the slope but were constant deep inside). These effects are one of the main limitations for the accuracy of distributed hydraulic systems, preventing them from having a wider use.

In order to simulate such an environment, the research was performed using a unique, specialistic dual-zone thermal chamber (Figure 11a), allowing the sensor specimen to be placed simultaneously in hot (up to +40 °C) and cold (up to −20 °C) zone. The temperature in both zones can be freely and independently controlled (Figure 11b) to reflect even the most extreme conditions of sensor operation.

The experiments were controlled using a wide range of independent temperature measurement techniques, installed both in the sensor’s body and in the space of the chambers in close proximity to the investigated sensor specimens. These techniques involved:distributed fiber optic sensors (DFOS) based on Rayleigh scattering, allowing for dynamic [23,24,25] measurements of temperature changes within the sensor body (Figure 12a) after activation of the temperature compensation system (TCS);distributed fiber optic sensors (DFOS) based on Raman scattering [26], allowing for static measurements of temperature in the chambers over the entire area of the frameworks (Figure 12b);a range of spot techniques: fiber Bragg gratings [27], conventional thermistors.

The first stage consisted of differentiating and stabilizing the temperature in chambers to establish the defined maximum difference over the length of the sensor. During this phase, the liquid in the tubes of TCS was stopped. Then, in the second stage, the liquid began to circulate in a closed circuit to transfer the heat along the sensor length and within its cross section, so finally, the temperature along the main tube was constant despite the maintained high differences in the external temperature fields. Optical fibers inside the sensor body registered temperature changes during this dynamic process with the frequency of 50 Hz. Figure 13 represents the example results by means of selected types of plots: in length (a), time (b), and both time and length (c, d) domains.

The temperature equalization process was performed with different initial parameters, including the presence and type of external thermal insulation (see also Figure 12a), arrangement (geometry) of TCS tubes or the speed of the liquid inside them. The laboratory research was supported by numerical simulations in a finite element (FE) method environment. The volume and nonlinear model of the sensor 8 m long involved the equation of heat transfer in a solid as well as three equations of liquid motion:Continuity equation expressing the principle of mass conservation;Equation describing the fluid velocity vector, resulting from Newton’s second law;Equation describing the principle of conservation of energy, expressed not directly by the internal energy but by the enthalpy of the volume element.

The FE model (Figure 14) was constructed to optimize laboratory research by determining the basic physical relationships and sensitivity of the system response to changes in the initial parameters. Numerical and laboratory results were consistent, showing that the most important parameter responsible for the effectiveness of the whole system was the speed of the liquid in TCS tubes, while the presence of an external layer of thermal insulation was of secondary importance. This was the reason why in the prototype version of the sensor, this insulation was removed (see Figure 5 and Figure 6). Another advantage of that approach was facilitation of the production, transportation, and installation process, due to the possibility of rolling the sensor in smaller diameter coils.

The second important insight was the ability to equalize the temperature over sensor length with an accuracy of ±0.5 °C for each 10 °C of initial difference, i.e., twice as good as originally expected. Hydraulic sensor specimens with lengths ranging from 5 to 50 m were analyzed during the tests.

## 4. In situ Application

### 4.1. General Description

Construction of embankments above the improved substrate is a challenging process requiring strict control, especially when ground improvement causes changes in soil parameters, which vary over time. Theoretical predictions, including even the most advanced calculations in three-dimensional FE models, are subject to considerable uncertainties. Thus, gathering reliable measurement data related to the technical condition of the structure are of great importance. They allow for improvement of the work’s quality but also reduce construction time and, thereby, save money. However, the most common measurements are those made using spot sensors and, thus, the analysis and technical assessment are considerably limited. The purpose of this section is to present the practical application of the new hydraulic sensor SHM Profiler, installed in field conditions to monitor settlements (vertical displacements) under an embankment during its construction.

The project is currently underway in Tuchów (southern Poland) by the general contractor—STRABAG company. The section is about 3 km long, and it is one of the most complicated ones being built in the Małopolska region (Lesser Poland), which influences the cost and work schedules. One of the main challenges is hydrological conditions, which pose the risk of water run-off due to the presence of a river and unfavorable topography. An innovative construction solution was chosen and applied within an experimental part of the embankment. Vertical filtration drains were installed to accelerate the consolidation of the ground layers under construction through effective water filtration.

Due to the complexity of the project and untypical construction solutions, it was reasonable to verify their effectiveness in practice by measurements of settlements (vertical displacements). It should be noted that traditional geodetic surveys performed at selected reference points (benchmarks) will not provide key information on displacements along the base of the embankment, i.e., within the area where, due to the highest loads, the expected displacements will also be the highest. The application of geodetic surveys only is usually not justified from an engineering and technical point of view (manual measurements, low accuracy, difficult access to benchmarks during operation); therefore, they should rather be treated as additional to other measurement methods.

### 4.2. Sensor Delivery, Location, and Installation

The road embankment in Tuchów was equipped with the new hydraulic sensor SHM Profiler after months of testing it in laboratory conditions. The decision was made to install the sensor in the transition zone between the substrate with vertical filtration drains and conventional construction (without the drains)—Figure 15a. This zone was selected for monitoring as a sensitive and crucial part of the overall structure. The sensor path was designed in such a way as to create a set of four sections (*A, B, C, D*), each at least 20 m long. Section *A* and *B* are located above the substrate with drains, while *C* and *D* above the native substrate without any improvements. In addition, a reference distributed fiber optic sensor (3DSensor [28]), dedicated to displacement measurements [29,30], was installed along the section A as shown in Figure 15b. Comparison between the results will allow for the assessment of the performance and accuracy of the new hydraulic measurement solution.

The sensors were delivered to the construction zones in coils (Figure 16a), enabling fast and convenient installation. It consisted of only unrolling the sensor and stabilizing it on the ground in the designed positions using the mounting brackets—Figure 16b. The hydraulic sensor was tied to the reference 3DSensor along section A to ensure the correct comparison between measured displacement values.

After sensor installation (Figure 17a), the field was ready for further construction (building and compacting the subsequent ground layers).

### 4.3. Measurement Sessions

All the sensor’s hydraulic tubes, pigtails, and signal wires were led to the technical well located at the embankment slope—Figure 17b (see also scheme in Figure 15a). A geodetic benchmark was also located inside the well, allowing displacements to be determined with reference to the global coordinate system. Due to the pilot character of the study, the measurements were performed periodically (in cyclic sessions). However, the technical infrastructure and the software were prepared for automatic monitoring over long-term. In such a case, it will be necessary to additionally secure the data-logger in an appropriate technical casing or container with access to a power supply (e.g., 230 V mains, rechargeable accumulators, solar panels).

Cyclic sessions were performed in accordance with the schedule of construction work. Elaborated connections were used to connect the sensor with the data-logger located inside the measurement station next to the slope and technical well—Figure 18a.

During measurements and data processing, there are a few important aspects that should be taken into consideration, including zero readings and correct identification of displacement directions. In the considered project, zero readings were taken after construction of the first ground layer of the embankment, with a thickness of approx. 100 cm. Weight of the soil mass caused the initial stabilization of the sensor and adjustment of the entire system. Calculated displacements with negative signs correspond to downward displacements (settlements)—Figure 18b. During measurements, the spatial resolution was set to approximately 100 cm. Any deviations in positioning the measurement points were corrected using an appropriate algorithm (see also line 8 in Table 1).

### 4.4. Example Results and Discussion

Graphs of height profiles estimated in subsequent measurement sessions (*P00, P01, P02,* and *P03*) along the entire length of the sensor, including all sections (*A, B, C, D*), are presented in Figure 19.

By performing multiple measurement sessions and using two independent pressure transducers (*a* and *b*), it was possible to analyze in detail the repeatability of the results. The interpretation of the determined differences between these two transducers within a single measurement session was carried out with reference to the maximum measurement range of the pressure transducers and assumed limit value of 1% of this range (error, which has been found to be acceptable)—Figure 20. The results indicate very good repeatability: the average error between the indications of the transducers was only 0.1% with a standard deviation of 0.2%. Consistence of the results showed that the pressures on both sides of the tube with the liquid separated by air bubble were measured correctly (the length of the air bubble was chosen appropriately).

The next and the most important step in data processing is calculation of displacement profiles as a difference between compensated height profiles measured in subsequent sessions after zero reading. Figure 21 shows he comparison in displacement profiles indicated by hydraulic SHM Profiler and reference distributed fiber optic 3DSensor, which was read with extremely high spatial resolution equal to 10 mm. The left axis corresponds to calculated displacements, while the right one to relative errors calculated in reference to the full scale of the applied pressure transducers (maximum measurement range of displacements). An acceptable error was assumed at 1% of the full scale, the same as in the case of the reputability analysis.

The example error analysis was done for the differences between the session *P02* and *P00*, i.e., when the ground layer with the thickness of approx. 3 m was constructed after zero reading. The mean relative error was 0.0% with a standard deviation of 0.1%. The mean absolute error was equal to 1.0 mm with a standard deviation of 3.1 mm. The limit values were not exceeded at any of the measuring points.

It is also worth noticing that during measurement sessions, the effectiveness of the thermal compensation system was also verified in selected checkpoints using spot thermistors. Temperature was constant inside the sensor body despite the differences in external temperatures, whose distributions were measured along the entire length of section A by distributed fiber optic sensing based on the Raman technique. Figure 22. shows the observed temperature distributions during sessions *P00* and *P02*. The maximum differences caused by the geometry of the embankment exceeded 10 °C in session *P02*—such conditions were previously simulated in a dual-zone thermal chamber. Equalizing the temperature along the length allowed for application of elaborated data processing algorithms and to achieve required accuracy of displacement measurements.

## 5. Conclusions

This article presents a new approach for structural health monitoring based on the application of a hydraulic sensor and system for vertical displacement measurements. The elaborated measuring tool could be operated automatically and remotely, and because of a specially designed system for thermal compensation, it is insensitive to temperature changes.

Design and performance of the sensor was checked based on a wide range of laboratory tests and three in situ installations, one of which was described in more detail in the article. The hydraulic sensor SHM Profiler is a solution dedicated especially for geotechnical structures, such as embankments, but also dams, pipelines, roads, and landslide or mining areas. Distributed measurements of settlements inside the embankment, constructed above the improved substrate with vertical filtration drains, allowed for its technical assessment and will be used during long term monitoring. Application of this type of measurement solution is also characterized by the following advantages:Shortening of the duration of construction works by early information about compaction level;Possible quality improvement of the works carried out by subcontractors,Verification of design assumptions and theoretical simplifications,Optimization of technological solutions applied in the future for similar projects,Identification of risks and possible reasons for failures,Providing objective documentation in case of warranty disputes,Improving the positive image and increasing trust in the general contractor.

Efficiency and performance of proposed solution was checked through theoretical (analytical) analysis, numerical (FEA) simulations, laboratory tests, and in situ installations. It is worth noticing that measurement sessions were carried out at the site during construction, including the presence of heavy construction equipment. Such difficult field conditions and loadings could be even higher than those predicted during normal operation of the structure.

Based on the comprehensive research, technical specifications of the hydraulic sensor were estimated and summarized in Table 3.

The presented solution is very promising in the context of long-term structural health monitoring of geotechnical structures, due to the possibility of automation (early-warning system) and creation of economical systems with very good *price-to-quality* ratio. Research is now continuing within three field applications, and lessons learned will allow for further improvements and a wider use.

## 6. Patents

Polish patent (PL), Application number: P.404271, Patent number: Pat.223638, Title: Method for multiple, automatic, and maintenance-free measuring of vertical displacement profile and a device for carrying out the method, Application date: 11 June 2013, Applicant: SHM System Sp. z o.o., Sp. komandytowa.

Polish patent (PL), Application number: P.430425, Title: Method and device for maintenance-free measurement of vertical displacement profile, Application date: 28 June 2019, Applicant: SHM System Sp. z o.o., Sp. komandytowa.

## Figures and Tables

**Figure 1 sensors-21-04678-f001:**
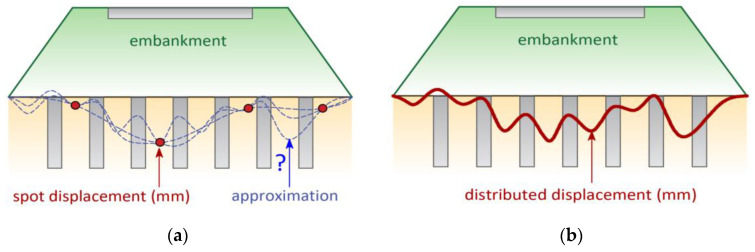
Idea of embankment settlement measurements using: (**a**) spot technique; (**b**) distributed technique.

**Figure 2 sensors-21-04678-f002:**
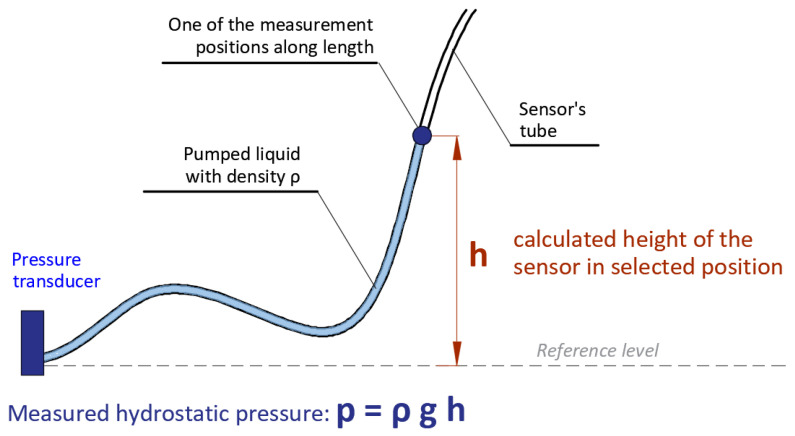
Basic operating principle for determining height by measuring hydrostatic pressure of the liquid column in selected position of the tube.

**Figure 3 sensors-21-04678-f003:**
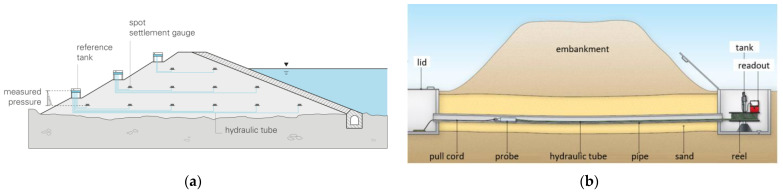
(**a**) Example of hydraulic measurement system based on multiple spot sensors [20]; (**b**) example hydraulic measurement system based on a single sensor moved by hand to different measuring locations [21].

**Figure 4 sensors-21-04678-f004:**
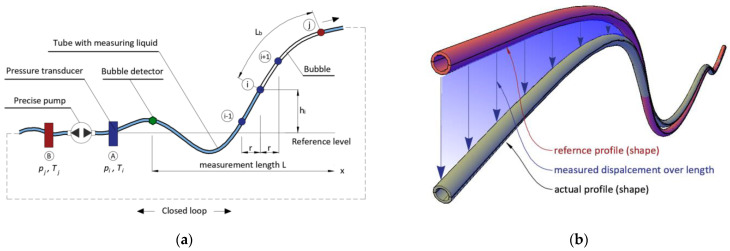
(**a**) Scheme of the new hydraulic system with closed loop; (**b**) principle of determining vertical displacements from measured profiles in subsequent measurement sessions.

**Figure 5 sensors-21-04678-f005:**
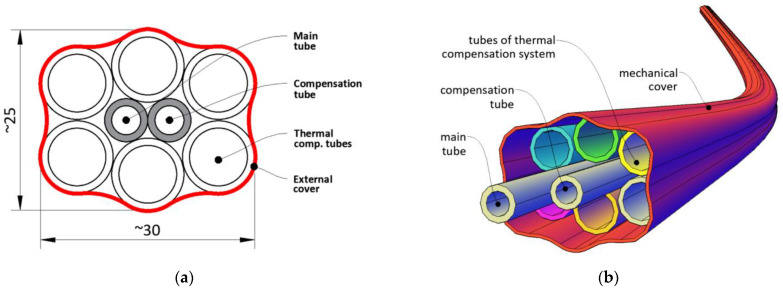
Design of new hydraulic sensor SHM Profiler: (**a**) cross section; (**b**) spatial visualization.

**Figure 6 sensors-21-04678-f006:**
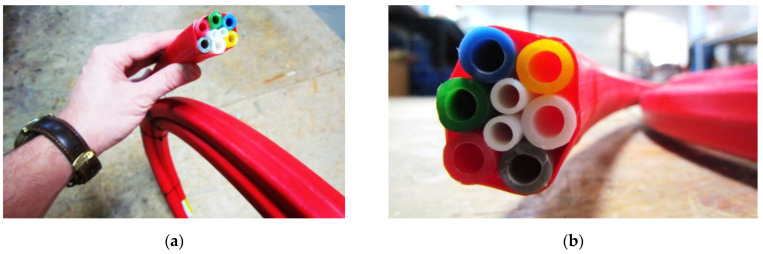
Prototype version of new hydraulic sensor SHM Profiler: (**a**) general view; (**b**) close-up to the cross section.

**Figure 7 sensors-21-04678-f007:**
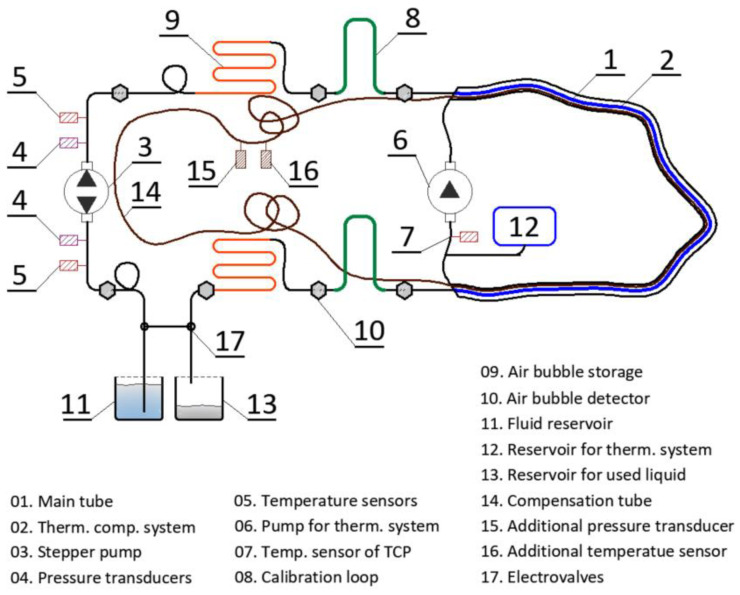
Scheme of hydraulic system designed for automatic operation of the new sensor.

**Figure 8 sensors-21-04678-f008:**
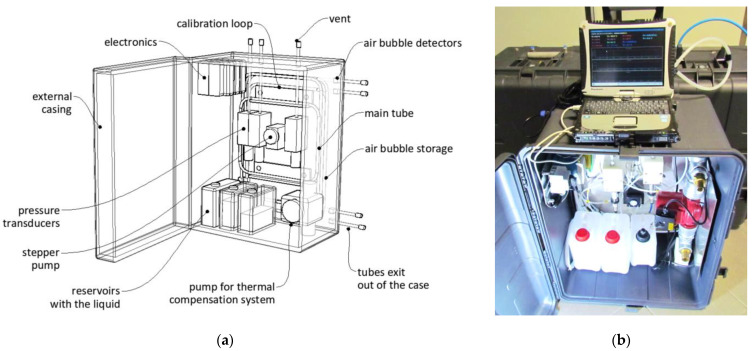
Hydraulic system designed for automatic operation of new sensor: (**a**) special visualization with key components; (**b**) the view of the case applied within filled measurements.

**Figure 9 sensors-21-04678-f009:**
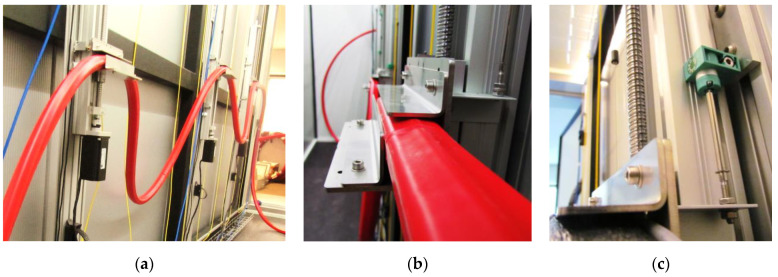
(**a**) View of the measurement station with stiff framework; (**b**) close-up to the mounting block of the sensor; (**c**) close-up to the one of inductive LVDT sensors applied during research.

**Figure 10 sensors-21-04678-f010:**
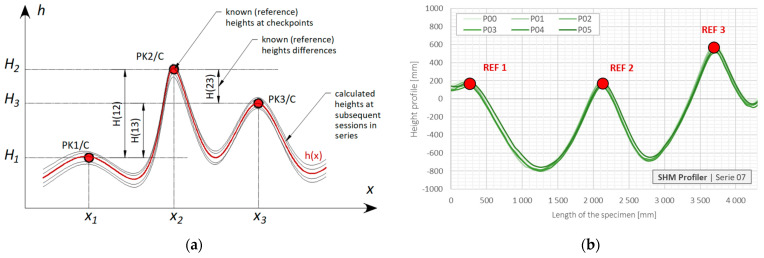
Analysis of sensor’s accuracy: (**a**) theoretical scheme; (**b**) graphical representation of example uncorrected results from one series of measurement sessions with checkpoints marked.

**Figure 11 sensors-21-04678-f011:**
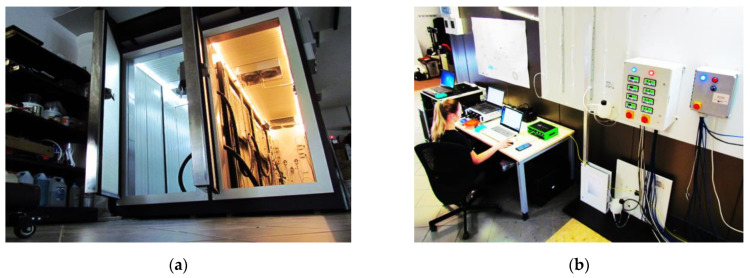
(**a**) Front view of dual-zone thermal chamber (with doors opened); (**b**) a stand for control and operating distributed fiber optic sensing and other measurement techniques during experiments.

**Figure 12 sensors-21-04678-f012:**
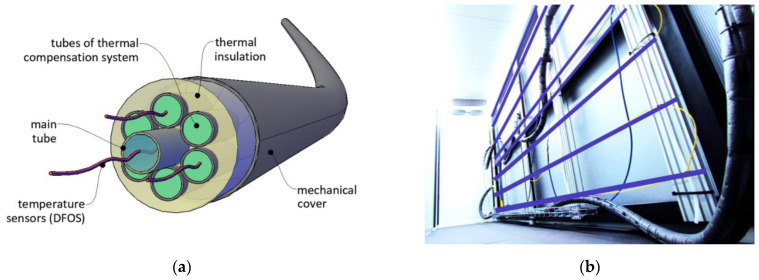
(**a**) Distributed optical fibers installed within the sensor body for dynamic measurements; (**b**) view of the framework inside the cold chamber equipped with distributed fiber optic sensors for static temperature measurements.

**Figure 13 sensors-21-04678-f013:**
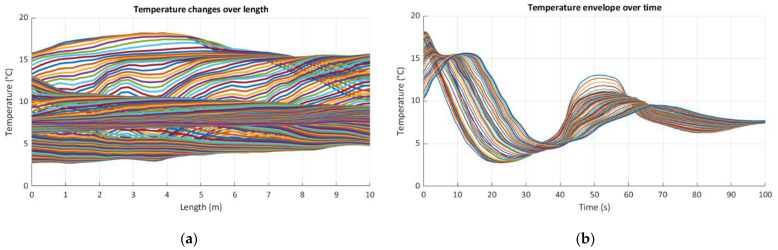
Example dynamic DFOS results showing the process of equalizing the temperature along the sensor: (**a**) in length domain; (**b**) in time domain; (**c**) spatial visualization in both time and length domains; (**d**) temperature map.

**Figure 14 sensors-21-04678-f014:**
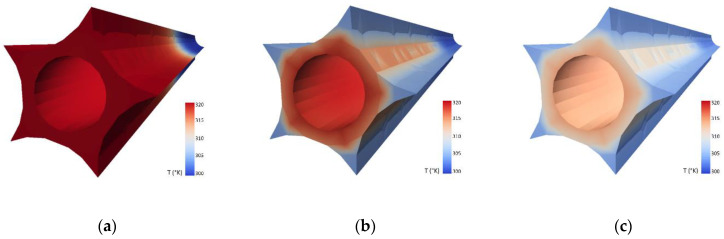
Example results of FE simulations presenting the temperature field along the sensor during equalization process after: (**a**) 0; (**b**) 10; (**c**) 30 s. of liquid circulating (the TCS tubes were removed to keep the appropriate clarity).

**Figure 15 sensors-21-04678-f015:**
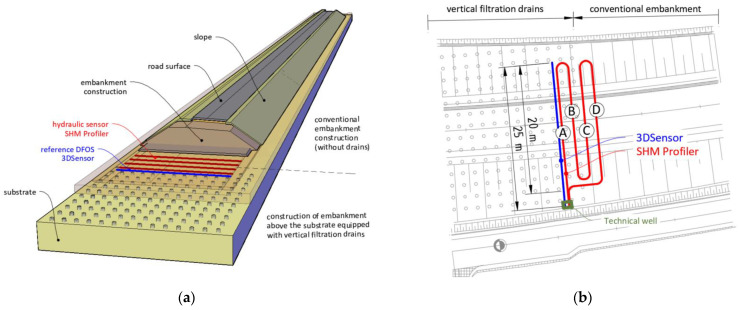
(**a**) Spatial visualization of embankment above the improved substrate; (**b**) location of the sensors, including four sections of hydraulic SHM Profiler and one section of distributed fiber optic 3DSensor (reference).

**Figure 16 sensors-21-04678-f016:**
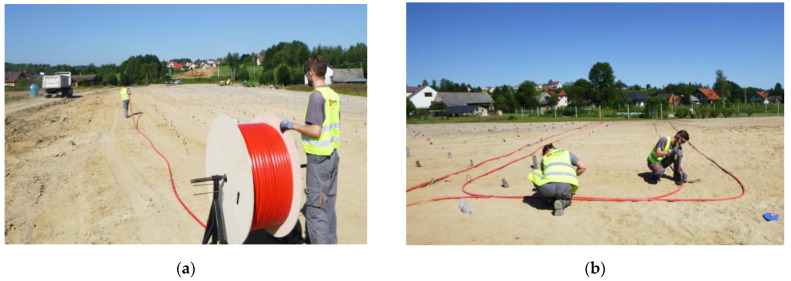
(**a**) Hydraulic sensor delivered to construction site in coil; (**b**) installation by stabilizing the sensor in the designed positions by means of the mounting brackets.

**Figure 17 sensors-21-04678-f017:**
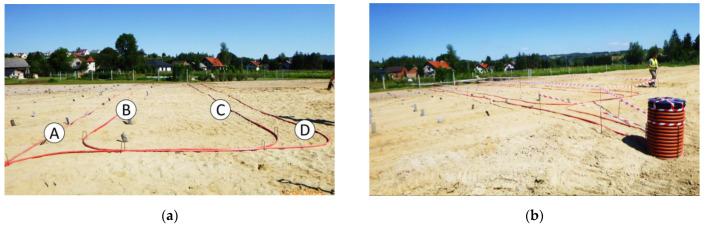
(**a**) The view of the construction field after installation of all the sensors; (**b**) the view of technical well protecting hydraulic tubes, pigtails, and signal wires of the sensors.

**Figure 18 sensors-21-04678-f018:**
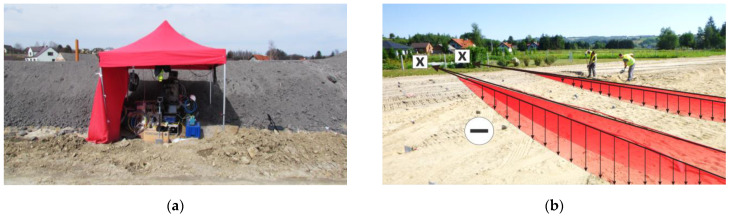
(**a**) The view of station next to the slope of embankment during one of the measurement sessions; (**b**) visualization of predicted displacement profiles with assumed signing (negative values correspond to downward displacements).

**Figure 19 sensors-21-04678-f019:**
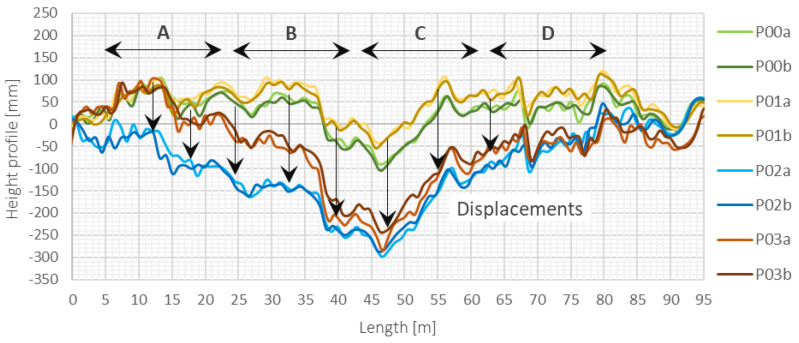
Height profiles estimated using the measurements of two pressure transducers (*a, b*) within hydraulic sensor during subsequent measurement sessions (*P00–P04*).

**Figure 20 sensors-21-04678-f020:**
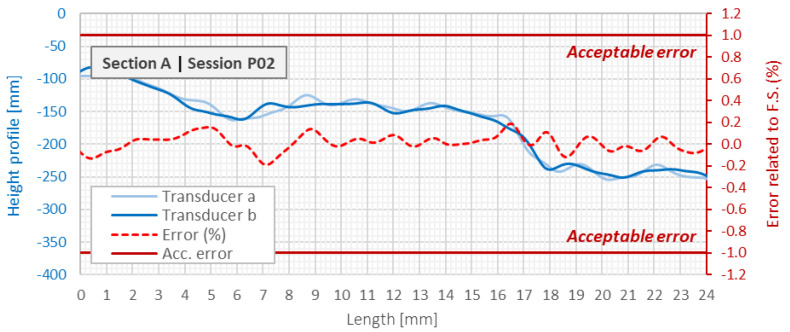
Reparability analysis between pressure transducer *a* and *b* during single measurement session *P02* along section *A*.

**Figure 21 sensors-21-04678-f021:**
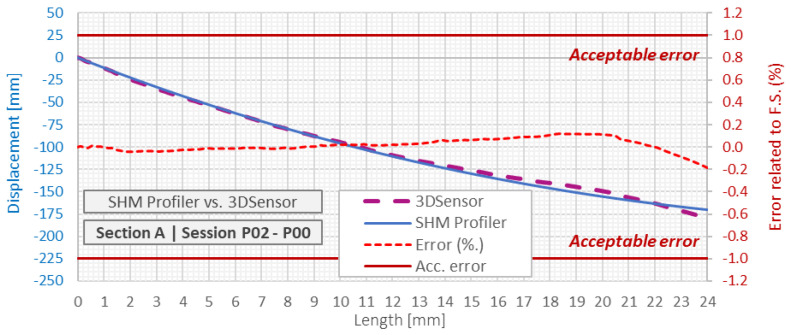
Displacements profiles indicated by hydraulic SHM profiler and DFOS 3DSensor with error analysis relative to the full scale of pressure transducers.

**Figure 22 sensors-21-04678-f022:**
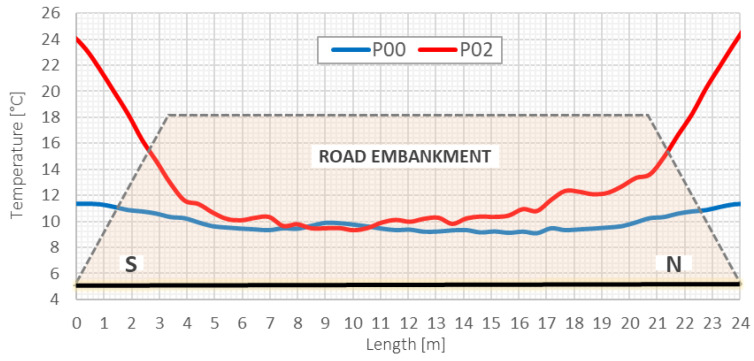
Temperature distributions along section *A* in session *P00* and *P02*, measured using the 3DSensor and DFOS technique based on Raman scattering (width and height of embankment are drawn on a different scale for clear presentation).

**Table 1 sensors-21-04678-t001:** Issues addressed during the development of new hydraulic and distributed sensor.

No	Issue	Work Description	Solution/Result
1	material of the tube, internal diameter of the tube, type (parameters) of liquid	statistical laboratory research on different materials, diameters, and liquids	selection of the tube’s material, internal diameter, and type of the liquid
2	long-term operation and aging of the system’s components	aging chamber tests and measurements during specified number of cycles	proved negligible influence of aging in predicted operation time
3	the minimum length of air bubble not influencing the pressures measured by two transducers on both sides	statistical laboratory research including pressure measurements with different lengths of air bubble at specified heights	empirical equation describing the relation between the max. measurement range and the required length of air bubble
4	maximum aeration of the liquid allowing for correct measurements	statistical laboratory research on performance of the liquids with different level of aeration measured by dedicated probe	application of air bubble detectors and elaboration of the procedure of automatic exchange of the liquid
5	the constant temperature along the sensor’s length for correct thermal compensation	verification of the effectiveness of the various systems for temperature equalization within dual-zone thermal chamber—see Section 3.2	application of special system involving external tubes around the main measuring and compensating tubes—see Section 2.3
6	thermal compensation due to the temperature changes between subsequent sessions	analysis of the state-of-the-art, analytical analysis and mathematical models checked under laboratory conditions	set of algorithms for thermal compensation; application of additional compensating tube—see Section 2.3
7	numerical model for optimization of the elements dedicated for thermal compensation	simulations using finite element method; nonlinear model based on volume elements and heat transfer equations	possible removal of thermal insulation, support and confirmation of results obtained from laboratory tests
8	compensation of the positioning error resulting from the limited accuracy of applied stepper pumps and actual geometry of the sensor	theoretical analysis involving mathematical–physical model with local approximation, confirmed by laboratory tests	algorithm for estimation displacements at any reference point, regardless of the actual measurement points
9	real accuracy of displacement profile measurements	statistical laboratory research under the control of reference, independent techniques for displacement measurements—see Section 3.1	data sheet and technical specifications of the new sensor
10	connection of the sensor’s segments to avoid destruction of air bubble	statistical laboratory research on different types of hydraulic connectors and methods of cutting off the tube face	elaboration of the optimum way to connect the sensor
11	data acquisition system	collaboration of interdisciplinary specialists to create electronic-information–hydraulic system	devices and software allowed to perform measurements remotely, automatically and without operator
12	verification of sensor performance in harsh environmental conditions	sensors’ demonstration installations in field conditions (one bridge and two geotechnical structures—embankments)	installation, measurements, data processing, real performance → lessons learned (see also Section 4)

**Table 2 sensors-21-04678-t002:** Relative ^1^ errors (%) calculated for uncorrected and corrected ^2^ data in series no. 7.

Session	Data Uncorrected			Data Corrected		
	h(12)	h(13)	h(23)	h(12)	h(13)	h(23)
S00	0.4	1.1	1.6	0.1	0.0	0.1
S01	0.4	0.9	1.4	0.2	0.2	0.3
S02	0.3	1.1	1.7	0.0	0.1	0.1
S03	0.4	1.1	1.6	0.1	0.0	0.0
S04	0.5	1.1	1.5	0.1	0.1	0.1
S05	0.5	1.1	1.5	0.2	0.3	0.1
Mean	0.4	1.1	1.6	0.1	0.1	0.1
Stdv ^3^	0.1	0.1	0.1	0.1	0.1	0.1

^1^ Error relative to the full displacement scale. ^2^ Correction involved both the constant error as well as the positioning error resulting from the limited accuracy of applied stepper pump (see also line 8 in Table 1). ^3^ Standard deviation.

**Table 3 sensors-21-04678-t003:** Technical specifications of hydraulic sensor for vertical displacement measurements.

Parameter	Value
Range of displacement measurement	up to 9 m
Accuracy of displacement measurement	<1.0% f.s.
Resolution of displacement measurement	±1 mm
Spatial resolution ^1^	>5 cm
Operation temperature ^2^	from −20 °C to +60 °C
Cross section dimensions	25 × 30 mm
Measurement length	up to 300 m
Weight of the sensor	0.5 kg/m

^1^ The lower the spatial resolution, the longer the measurement time. ^2^ Integrated thermal compensation system.

## Data Availability

The data presented in this study are available on request from the corresponding author.

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
