# Peer review of "New Hydraulic Sensor for Distributed and Automated Displacement Measurements with Temperature Compensation System"

_sensors, 2021, doi:10.3390/s21144678_

Round 1

Reviewer 1 Report

This paper describes development of distributed sensor system and its incorporation into structural health monitoring (SHM) systems for non-homogenous media.  The paper is on a very important topic and will contribute to development of new sensors and approach to monitoring.  The new system seems very practical and shown to be effective through a set of laboratory and field measurement. Authors are encouraged to address the comments below to improve their work and raise the quality for inclusion in the prestigious MDPI Sensors journal.

Editorial Comments;

  • The paper is written well. However, there are still grammatical and editorial issues to address and needs to be reviewed accordingly.
  • Example- Lines 72-73- comma is not needed after “promising.”
  • Example- Line 84- “acquisition” should read “data acquisition.”
  • Example- Lines 89-91- “Distributed hydraulic …” does not read well.
  • Exampe- Lines 105-106- “Systems” should be “System.”
  •  
  • Line 233- “focused of “ should read “focused on.”
  •  
  • Line 368- What does “Investment” mean?

Technical Comments;

  • Line 89-99- Description of the hydraulic distributed sensor system and measurement process is not clear enough. For example, how are the pressure measured at certain points, etc. Need better and wider explanation of existing distributed sensor systems. Without a clear description, the disadvantages described in Lines 103 to 116 cannot be understood.
  • For example, why does the temperature influence the results (Line 108)?
  • Section 2.1- Basic operation principle- This section does not clearly and adequately describe how the system works. E.g.,
    • Does the bubble move along the tube?
    • Do pressure gages move with the bubble along the tube?
    • How is the location and integrity of bubble monitored inside the tube?
    • Is there any pressure sensor along the main tube? Figure 7 does not show any pressure sensor along the main tube.
    • How is the pressure at one location along the tube measured?
    • What is the pressure sensor? What is the role of air bubble? How do the sensors move with air bubble movement?

Author Response

Dear Reviewer,

thank you for all your comments and suggestions. Please find attached our responses and explanations. Also, the text of the manuscript was improved by additional descriptions.

Yours faithfully, Tomasz Howiacki

Reviewer 2 Report

Research is very interesting. It is curious to know the applicability of developing device. How deep can we put this sensor? Are there any section deformation of the tube? But I understand that you show only some steps of your study. I found only one remark - it is figure 3, it has very poor resolution, it is impossible to read text.  I would recommend to publish this work!

Author Response

Dear Reviewer,

thank you for your positive comment. Please find attached our response. Also, the text of the manuscript was improved by additional descriptions.

Yours faithfully,
Tomasz Howiacki

Round 2

Reviewer 1 Report

Thank you for your responses and good clarifications. 

I still have few questions that I believe any reader may have the same.

1- When the pump is stopped for taking pressure reading, how does the system know where exactly the bubble is?

2- Although described that the length of bubble is found experimentally to avoid the effect of one water column on the other, it is not clear how this can be accomplished. For example, with the position of bubble in Figure 4a, the pressure of water column on the right of bubble will affect the pressure of the point at the left end  of the bubble. Additionally, the bubble can be pressurized with such arrangement and its pressure can be added to the hydraulic pressure at either end of the bubble. The pressure in the air bubble will vary with the position of the bubble. This needs to be clarified and explained.

3- Although, the hydrostatic pressure formula makes the basis of the measurement, it is not clear whether a more complex formulation is used for accounting the bubble pressure or not.

Author Response

Dear Reviewer,

please find attached our further explanations.

Yours faithfully,
Tomasz Howiacki
